# The Pro-Oncogenic Protein IF_1_ Promotes Proliferation of Anoxic Cancer Cells during Re-Oxygenation

**DOI:** 10.3390/ijms241914624

**Published:** 2023-09-27

**Authors:** Riccardo Righetti, Silvia Grillini, Valentina Del Dotto, Anna Costanzini, Francesca Liuzzi, Claudia Zanna, Gianluca Sgarbi, Giancarlo Solaini, Alessandra Baracca

**Affiliations:** 1Laboratory of Biochemistry and Mitochondrial Pathophysiology, Department of Biomedical and Neuromotor Sciences, University of Bologna, 40126 Bologna, Italy; riccardo.righetti3@unibo.it (R.R.); silvia.grillini3@unibo.it (S.G.); valentina.deldotto2@unibo.it (V.D.D.); claudia.zanna@unibo.it (C.Z.); gianluca.sgarbi@unibo.it (G.S.); alessandra.baracca@unibo.it (A.B.); 2Department of Translational Medicine, St. Anna University Hospital, University of Ferrara, 44124 Ferrara, Italy; anna.costanzini@unife.it; 3Department of Medical and Surgical Sciences Maternal-Infantile and Adult, University of Modena and Reggio-Emilia, 41125 Modena, Italy; francesca.liuzzi@unimore.it

**Keywords:** anoxia, mitochondria, ATP synthase, oxidative phosphorylation, metabolism, autophagy, mitophagy, biogenesis, 143B osteosarcoma cells, HCT116 colon carcinoma cells

## Abstract

Cancer cells overexpress IF_1_, the endogenous protein that inhibits the hydrolytic activity of ATP synthase when mitochondrial membrane potential (Δμ_H_^+^) falls, as in ischemia. Other roles have been ascribed to IF_1_, but the associated molecular mechanisms are still under debate. We investigated the ability of IF_1_ to promote survival and proliferation in osteosarcoma and colon carcinoma cells exposed to conditions mimicking ischemia and reperfusion, as occurs in vivo, particularly in solid tumors. IF_1_-silenced and parental cells were exposed to the FCCP uncoupler to collapse Δμ_H_^+^ and the bioenergetics of cell models were validated. All the uncoupled cells preserved mitochondrial mass, but the implemented mechanisms differed in IF_1_-expressing and IF_1_-silenced cells. Indeed, the membrane potential collapse and the energy charge preservation allowed an increase in both mitophagy and mitochondrial biogenesis in IF_1_-expressing cells only. Interestingly, the presence of IF_1_ also conferred a proliferative advantage to cells highly dependent on oxidative phosphorylation when the uncoupler was washed out, mimicking cell re-oxygenation. Overall, our results indicate that IF_1_, by allowing energy preservation and promoting mitochondrial renewal, can favor proliferation of anoxic cells and tumor growth. Therefore, hindering the action of IF_1_ may be promising for the therapy of tumors that rely on oxidative phosphorylation for energy production.

## 1. Introduction

Mitochondria play essential roles in energy production, redox homeostasis, cellular metabolism, cell signaling, and cell death. The function of mitochondria depends on their structure and dynamics, and mitochondrial activities are crucial in cancer cells to sustain proliferation, cope with severe stress conditions such as hypoxia and anoxia, and spread as metastasis [1,2,3,4,5,6,7]. Despite the genetic and metabolic heterogeneity between cancer types and stages, tumor cells share common features, including the high energy requirement and the ability to overcome apoptosis [8,9,10]. Programmed cell death is strictly regulated by the structural integrity and functionality of mitochondria [11], which is primarily defined by the inner membrane electric potential (Δψ_m_), the main component of the electrochemical proton gradient (Δμ_H_^+^) coupled to the synthesis of ATP by ATP synthase (F_1_F_o_-ATPase) [12,13]. Under conditions of severe oxygen deficiency, the mitochondrial electrochemical gradient collapses and the enzyme is forced to work in reverse, hydrolyzing ATP to restore the Δμ_H_^+^. However, excessive ATP waste is prevented by IF_1_, an endogenous inhibitor of F_1_F_o_-ATPase hydrolytic activity [14,15,16]. IF_1_ is activated as a dimer by both mitochondrial membrane potential decrease and matrix acidification and it binds to the catalytic portion of the ATP synthase, F_1_, inhibiting its ATPase activity to avoid energy dissipation [17]. The function of this small evolutionarily conserved protein is well established in normal cells where IF_1_ attains a good balance between cellular energy charge and the Δμ_H_^+^ [18]. This allows cells to escape death as Δμ_H_^+^ and intracellular ATP are kept near their steady-state values, and only transient and limited changes can be safely induced by endogenous events or by interactions with the environment [19]. Prolonged or large perturbations in either Δμ_H_^+^ or ATP level impair metabolic and ion homeostasis, which can lead to cell death [19]. Therefore, dysfunctional mitochondria identified by aberrant levels of Δμ_H_^+^ (or Δψ_m_) must be removed from cells, and this requires efficient mitochondrial quality control mechanisms. Among the several mechanisms acting at the organelle level to achieve mitochondrial quality control, mitophagy, a selective type of autophagy, is particularly efficient in removing unhealthy mitochondria [20,21,22,23]. Three canonical and well-characterized mitophagy pathways occur upon autophagosome recognition of the different receptors in a damaged mitochondrion [24]. PINK1 (PTEN-induced putative kinase 1)/Parkin-mediated ubiquitination of outer mitochondrial membrane proteins enables the autophagy cargo receptors p62 to bridge the interaction between mitochondria and autophagosomes [25]. Alternatively, BNIP3 (BCL2/adenovirus E1B 19 kDa protein-interacting protein 3) and FUNDC1 (FUN14 Domain Containing 1) are two proteins localised in the outer mitochondrial membrane (OMM) that can directly bind the microtubule-associated protein light chain 3 active fragment (LC3-II) conjugated to autophagosome membranes through a mechanism modulated by their phosphorylation status [26,27,28]. Unhealthy mitochondria are sequestrated and degraded in the autophago-lysosomes, avoiding the release of apoptotic factors and preventing the cascade of cell death [29,30]. The proteins involved in the mitophagic processes are dysregulated in cancer patients [24], but tumorigenesis could be promoted or not by mitophagy, depending on the pathway involved, the type of tumor and possibly on the stage of tumor development [31,32]. It is becoming clear that mitophagy pathways are intricately linked to the metabolic rewiring of cancer cells by providing components to produce energy and build up macromolecules to support high bioenergetic demand and rapid growth of the tumor [33].

Indeed, the mechanisms of action of IF_1_ in cancer cells are still a matter of discussion and include apoptosis escape [11,34], promotion of autophagy [35], regulation of calcium homeostasis [36], protection from anoxia and near-anoxia death [37,38], and metabolism reprogramming [39], and others [40].

Considering the IF_1_ overexpression and the poorly investigated effect of IF_1_ in tumor cells exposed to oxygen deficiency [37,41,42], we hypothesized that IF_1_ may play a role not only in protecting cells from death, but also in promoting proliferation of re-oxygenated cells. In two different models of cancer cells exposed to uncoupling conditions that simulate the main features of anoxia-induced bioenergetic dysfunctions (henceforth anoxia-mimicking conditions), we investigated whether IF_1_ by modulating mitophagy could promote cell proliferation when the FCCP uncoupler was removed, allowing normal functioning of oxidative phosphorylation (OXPHOS). The biochemical changes caused in cells by exposure to an uncoupler only partially overlap with those caused by oxygen deprivation. However, due to the significant technical difficulties in measuring functional parameters in anoxic cells, to perform this study we chose to expose the cells to the uncoupler FCCP that promotes the collapse of mitochondrial membrane potential and the relative decrease in the cellular energy charge, as occurs in anoxia [43,44].

Targeting quality control mechanisms is now considered a promising therapeutic approach for cancer treatment, and research efforts over the past decade have been directed at identifying drugs that can selectively modulate mitophagy in cancer cells.

Here, we show that IF_1_ significantly contributes to the modulation of mitochondrial homeostasis in anoxic cancer cells and provides a great advantage to OXPHOS-dependent cells during re-oxygenation.

## 2. Results

### 2.1. Cell Growth and Mitochondrial Mass of Both Osteosarcoma 143B and Colon Carcinoma HCT116 Cell Lines

The role of IF_1_ in the modulation of molecular mechanisms and signaling that promotes (i) the adaptation and survival of cancer cells to a stressful condition such as anoxia and (ii) their proliferation upon re-oxygenation was investigated. To this aim, the study was performed by using cancer cell lines derived from different human tumors: the osteosarcoma 143B, and colon carcinoma HCT116 cell lines. The two cancer cell types show a similar proliferation rate when grown in high glucose medium for 72 h (Figure 1A). Notably, the HCT116 cells have a higher content of mitochondria compared with the 143B cells, as established by the quantification of mitochondrial proteins such as VDAC1 and TOM20 (Figure 1B,C). Furthermore, electrophoretic analysis of cell lysates followed by Western blotting and immunodetection revealed that both the IF_1_ protein level and the densitometric ratio of the IF_1_ band to that of the ATP synthase d subunit are significantly higher in HCT116 cells than in 143B cells. Indeed, the ratio is nearly 6-fold higher in HCT116 than in 143B cells (Figure 1D,E).

### 2.2. IF_1_ Preserves the Energy Charge of Cancer Cells under Uncoupling Conditions

To investigate the role of IF_1_ in modulating the functional state of mitochondria in osteosarcoma and colon carcinoma cells adapted to anoxic conditions, stably IF_1_-silenced clones were obtained from both 143B and HCT116 cells. The IF_1_ content of the clones used in this study was less than 3 and 9% of the parental osteosarcoma and colon carcinoma cells, respectively (Appendix A). Adaptation of cancer cell bioenergetics to anoxia was achieved by exposing all cell types to the uncoupler FCCP that abolishes the electrochemical proton gradient impeding the occurrence of OXPHOS. All the cell types were loaded with TMRM and Δψ_m_ changes induced by the addition of FCCP were dynamically analyzed by flow cytometry (Figure 2A,B). An instantaneous, steep and progressive decrease in Δψ_m_ occurred independently of the IF_1_ expression. However, only in IF_1_-expressing cells the Δψ_m_ reached a very low steady state in a short time frame (approximately 10–15 min). Indeed, the mitochondrial membrane potential was partially preserved in IF_1_-silenced cells by the ATP synthase working in reverse, as shown by the oligomycin-induced Δψ_m_ collapse (Figure 2A,B). The effect of IF_1_ on the uncoupler-induced Δψ_m_ deccrease was confirmed by fluorescence microscopy analysis in TMRM-loaded cells exposed to FCCP for up to 24 h (Figure 2C,D). Our findings show that uncoupling treatment triggered a sudden perturbation in cancer cells that rapidly reached a new steady state strictly dependent on the presence of IF_1_ and maintained this state for at least 24 h. Concurrently, the IF_1_ expression affects the energy charge of cancer cells grown under uncoupling conditions. Indeed, under these conditions, the IF_1_-silenced cells showed a significantly higher (about three-fold) ADP/ATP ratio compared with the related parental cells (Figure 3). Remarkably, the energy charge of the osteosarcoma cells was found to be less sensitive to uncoupling than that of colon carcinoma cells. In particular, uncoupling did not affect the ADP/ATP ratio of osteosarcoma control cells while the ratio was approximately three-fold and more than fifteen-fold increased in uncoupled IF1-silenced cells derived from osteosarcoma and colon carcinoma, respectively (Figure 3). These results, together with the lower mitochondrial mass of the 143B compared with HCT116 cells, strongly suggest that the biochemical phenotype of the two cancer cell types is different, with the HCT116 cells being more prone to produce ATP via OXPHOS than the 143B cells. The proliferative capacity of the osteosarcoma and colon carcinoma controls in high glucose medium was significantly decreased after 24 h FCCP exposure (Figure 4A,B). The increase in dead cells upon the exposure to the uncoupler was about 5% for all osteosarcoma-derived cells. IF_1_-silenced colon carcinoma cells exhibited a slightly greater increase in cell death compared with the controls (approximately 12%) (Figure 4A–D). Notably, under anoxia-mimicking conditions, proliferation of IF_1_-silenced osteosarcoma cells appeared to be arrested, while a significant increase in death was observed in IF_1_-silenced colon carcinoma cells (Figure 4A,B).

### 2.3. Uncoupling Promotes Mitophagy in Cancer Cells

Cancer cells commonly display a high energy demand to sustain their high proliferation rate and invasiveness. Although IF_1_ provides an advantage to uncoupled cancer cells by allowing energy charge preservation while limiting its decrease, it is widely described that abolishment of Δψ_m_ across the inner mitochondrial membrane (IMM) could trigger mitophagy, a selective autophagic degradation of mitochondria. Considering the different Δψ_m_ values observed in IF_1_-silenced and control cells, we hypothesized a dissimilar activation of the quality control mechanism of mitochondria. To address this issue, markers of autophagic mitochondrial degradation (mitophagy) were assessed in all cell types after 24 h of culturing in the presence of FCCP. Indeed, we analyzed the cellular content of PINK1, whose accumulation in the outer mitochondrial membrane (OMM) following the collapse of Δψ_m_ triggers mitophagy [45,46], and LC3B, a member of the LC3 protein family whose incorporation represents the maturation step of autophagosomes [29,47]. Interestingly, the level of PINK1 was found to be increased approximately two- to three-fold in all uncoupled IF_1_-expressing cells. Conversely, the level of PINK1 was found to be significantly reduced (approximately 35%) or nearly unchanged in IF_1_-silenced cells derived from osteosarcoma and colon carcinoma, respectively (Figure 5). These results were further corroborated by evaluating the localization of mitochondria and autophagosomes in uncoupled IF_1_-expressing and IF_1_-silenced osteosarcoma cells. To this aim, osteosarcoma-derived clones stably expressing mtRFP, a red fluorescent protein targeted to mitochondria, transiently transfected with a plasmid expressing LC3 fused to the YFP, were analyzed by fluorescence microscopy. Colocalization of mitochondria (red) and autophagosomes (green) was assessed by acquiring and merging the two fluorescence signals (mtRFP and LC3-YFP) from the same optical field. Under basal conditions, mitophagy was scarcely active in both IF_1_-expressing and IF_1_-silenced cells, as indicated by the low occurrence of colocalization events (Figure 6A, right panels). Remarkably, both control and IF_1_-silenced cells displayed a substantial increase in mtRFP/LC3-YFP colocalization events after 24 h of deferoxamine treatment (DFO), proving that both cell types responded to mitophagic stimuli (Figure 6B, right panels). However, after 24 h of growth under anoxic-mimicking conditions, only the IF_1_-expressing cells exhibited a significant increase in mitophagy, as indicated by the higher number of colocalization events (Figure 6, right panels C and A). Unexpectedly, higher levels of the LC3B-II/I ratio were found in all uncoupled cancer cells compared with basal growth conditions (Figure 7). Indeed, this increase was greater in the uncoupled IF_1_-silenced cells than in the corresponding uncoupled controls and was likely due to autophagy activation by low cellular energy charge.

### 2.4. IF_1_ Does Not Affect the Level of Mitochondrial Mass in Uncoupled Cancer Cells

The activation of mitophagy in IF_1_-expressing cancer cells prompted us to evaluate whether mitochondrial mass changes have occurred in cells exposed to conditions that mimic anoxia. Indeed, we analyzed the cellular content of mitochondrial proteins such as VDAC1 and TOM20 in all cell types and no differences were observed after exposure to the uncoupler in either IF_1_-expressing or IF_1_-silenced cells (Figure 8). Since we did not observe any change in mitochondrial mass in uncoupled IF_1_-expressing cells and considering that it depends on the balance between mitophagy and biogenesis, we hypothesized that under uncoupling conditions, both processes could be activated. Therefore, we evaluated by both SDS-PAGE and immunodetection the levels of SIRT1 and PGC-1α, typical markers of mitochondrial biogenesis [48,49,50,51,52], in all cell types following exposure to FCCP (Figure 9). Interestingly, PGC1-α protein levels were increased (about 50%) in uncoupled osteosarcoma IF_1_-expressing cells compared with basal conditions, while protein levels decreased in uncoupled IF_1_-silenced cells. Similarly, uncoupled colon carcinoma IF_1_-expressing cells showed an increase in PGC1-α levels, although it did not reach statistical significance. Unexpectedly, SIRT1 levels were found almost unchanged in osteosarcoma IF_1_-expressing cells and significantly decreased in IF_1_-expressing colon carcinoma cells. All IF_1_-silenced cells showed slightly decreased SIRT1 levels.

### 2.5. IF_1_ Expression Favors the Proliferation of Uncoupled Cancer Cells When Re-Oxygenated

The protective role exerted by IF_1_ on the bioenergetics of cancer cells under anoxia-mimicking conditions led us to hypothesize that IF_1_ may affect cell proliferative capacity when re-oxygenation occurs. To address this issue, both osteosarcoma- and colon carcinoma-derived cells were seeded and grown for 96 h in high glucose medium after 24 h exposure to FCCP. To better simulate cell adaptation to anoxic conditions that physiologically include severe glucose shortage, all cell types were exposed to the uncoupler in the presence of 5 mM glucose. Under these conditions, the proliferative capacity of the osteosarcoma and colon carcinoma controls was significantly decreased (Figure 10A,B). The percentage of dead cells after exposure to the uncoupler was about 8% in all the osteosarcoma cells, whilst it was slightly higher in the colon carcinoma IF_1_-silenced cells (about 16%) compared with controls (about 5%) (Figure 10C,D). Notably, the low glucose availability highlighted and confirmed the tendency for only IF_1_-silenced colon carcinoma cells to die when the cells were grown under anoxia-mimicking conditions (Figure 10A,B).

Interestingly, when cells were re-seeded and grown for 96 h in high glucose medium after FCCP exposure, colon carcinoma-derived IF_1_-expressing cells showed a significantly greater proliferation rate than IF_1_-silenced cells, while no differences were observed between osteosarcoma cells expressing or not expressing IF_1_ (Figure 11A,B).

## 3. Discussion

The main finding of the present study is that the endogenous inhibitor of ATP synthase, IF_1_, is not only essential for maintaining energy charge in cancer cells exposed to an uncoupler, a condition mimicking anoxia, but it can also allow a prompt recovery of proliferation when the uncoupler is removed, a condition mimicking re-oxygenation. Furthermore, this study proposes molecular mechanisms adopted by the IF_1_-expressing cells during anoxia which include mitophagy to remove damaged mitochondria avoiding the activation of death pathways [53]. Importantly, the removal of dysfunctional mitochondria is counterbalanced by their biogenesis, resulting in the preservation of both mitochondrial mass and function in anoxic/uncoupled cancer cells. Consequently, functioning mitochondria can sustain proliferation of cancers cells upon re-oxygenation or when the uncoupler is removed. Finally, we present a model of ischemia-reperfusion injury in cancer cells that is useful for studying the bioenergetics of cells exposed to conditions of anoxia or near-anoxia followed by re-oxygenation. Therefore, it extends the model we have previously reported [37]; on the other hand, it is a cellular model that represents what happens in a re-oxygenated mammalian organ after ischemia [54,55,56,57]. Incidentally, the model reported here can complement an anoxic model of adult mouse cardiomyocytes for in vivo ischemia-reperfusion injury that was recently described [58].

The protection exerted by IF_1_ on the bioenergetics of anoxic/uncoupled cancer cells seems to be general. Indeed, it is independent of both the metabolic phenotype and IF_1_ level of cancer cells, with the IF_1_/ATP synthase d subunit densitometric ratio in HCT116 being nearly six-fold higher than that of 143B cells. However, the protective action of IF_1_ is more relevant to cancer cells that rely on OXPHOS to produce ATP rather than glycolysis as occurs in HCT116 compared with 143B cells. Indeed, HCT116 cells not only exhibit a greater mitochondrial mass, but their energy charge is also more responsive to the uncoupler-induced membrane potential collapse than that of 143B cells. Notably, the silencing of IF_1_ resulted in a 3- and 15-fold increase in the ADP/ATP ratio of the uncoupled 143B and HCT116 cells, respectively.

After exposure to conditions that mimic anoxia, the mitochondrial mass of all cancer cells is completely preserved independently of IF_1_ expression and related mitochondrial membrane potential. However, the volume of mitochondria in a cell results from the balance between two processes, biogenesis and mitophagy, whose rates can vary depending on the stress conditions [20,59,60]. Indeed, in uncoupled IF_1_-expressing cancer cells, only the maintenance of the mitochondrial mass results from the activation of mitophagy, and the qualitative and quantitative selective control of mitochondria is balanced by the biogenesis process. Increases in both PINK1 and LC3BII levels were observed in both osteosarcoma and colon carcinoma control cells. Furthermore, activation of mitophagy in IF_1_-expressing osteosarcoma cells was confirmed by fluorescence microscopy analysis, showing co-localization of mitochondria and autophagosomes. The occurrence of mitophagy in cells exposed to stress conditions has been previously reported in several studies, including some on closely related topics [24,47,61]. Campanella et al. analyzing HeLa cells maintained under normal culture conditions reported that IF_1_ appears to limit the generation of mitochondrial ROS, limiting the degradation of mitochondria by autophagy which is increased by IF_1_ knockdown [35]. Lefebvre et al. showed the importance of PARK2 for mitophagy, stating that in the absence of overexpressed PARK2, there was no significant mitophagy in U2OS osteosarcoma cells in response to CCCP [62]. In the experiments reported here, uncoupling of the inner mitochondrial membrane was induced in different cell lines by FCCP, a molecule analogous to CCCP. Nonetheless, we could observe mitophagy without manipulating the cells to overexpress PARK2. In another study, Malena et al. showed that mutant adenocarcinoma and rhabdomyosarcoma cells, which differ in their bioenergetic profile, resulted in different mitophagy responses to the energy stress caused by the mutations [63]. Thus, there is a broad consensus that mitophagy is upregulated in cancer cells stressed by energy deficiency and decreased membrane potential. The consequence of increased mitophagy should be a decrease in the mitochondrial volume of the cells. However, here we demonstrate that the removal of damaged and depolarized mitochondria was only activated in uncoupled IF_1_-expressing cells and was counterbalanced by the activation of signaling pathways leading to an increase in the level of the PGC1-α transcription factor. Both 143B and HCT116 cells exposed to anoxia-mimicking conditions for 24 h showed PGC1-α upregulation (Figure 9) and the same mitochondrial volume as cells grown in normoxia. Of note, all the IF_1_-silenced cells downregulated PGC1-α. These investigations showed the crucial role of IF_1_ in stimulating mitophagy and biogenesis and in maintaining their steady state under conditions mimicking anoxia. Indeed, renewal of mitochondrial mass is the result of both IF_1_-dependent mitochondrial membrane depolarization and maintenance of energy charge at a sufficiently high level. This statement is supported by the involvement of PINK1 in the activated mitophagic pathway in uncoupled cells expressing IF_1_ [25,64]. In addition, the inability of uncoupled IF_1_-silenced osteosarcoma cells to renew mitochondrial mass makes it highly unlikely that stimulation of mitophagy /biogenesis depends solely on cellular energy. Indeed, the glycolytic phenotype of osteosarcoma cells greatly limits the decrease in energy charge in IF_1_-silenced cells exposed to uncoupling conditions.

Overall, our study highlights that cancer cells under anoxia-mimicking conditions are favoured by their high IF_1_ content. The reported experiments also indicate that the molecular mechanisms supporting cancer cells resistance to anoxia affects proliferation following re-oxygenation in cells that rely on aerobic oxidative metabolism to produce energy. Indeed, upregulation of the mitochondrial turnover in IF_1_-expressing colon carcinoma cells, which exhibited both an OXPHOS-dependent biochemical phenotype and a high IF_1_/ATP synthase ratio, resulted in a higher proliferation rate upon re-oxygenation compared with IF_1_-silenced cells.

Our results summarized in Figure 12 shed light on the role played by the ATP synthase inhibitor IF_1_ in preventing growth arrest/death of anoxic cancer cells and in promoting proliferation of re-oxygenated OXPHOS-dependent cells. On this basis, we suggest considering IF_1_ as a possible target of anticancer drugs aimed at inhibiting its action on ATP synthase in order to slow down or arrest the growth of tumors, in particular those exhibiting high OXPHOS-dependent metabolism.

## 4. Materials and Methods

### 4.1. Cell Culture

Human osteosarcoma 143B cells and human colon carcinoma HCT116 cells were maintained at 37 °C with 5% CO_2_ in Dulbecco’s Modified Eagle Medium (DMEM) containing 10% Fetal Bovine Serum (FBS), 100 U/mL penicillin, 100 μg/mL streptomycin and 0.25 μg/mL amphotericin B. The culture medium was supplemented with growth substrates according to the ATCC (American Type Culture Collection) guidelines: 25 mM glucose, 4 mM glutamine and 1 mM pyruvate or 16.7 mM glucose and 2 mM glutamine for 143B and HCT116 cells, respectively.

Cell culture reagents were purchased from Gibco (Life Technologies Italia, Monza, Italy), with the exception of glucose and pyruvate provided by Sigma-Aldrich (Merck Life Science S.r.l., Milan, Italy. All the experiments were performed by seeding cells in complete medium containing 25 mM glucose, 4 mM glutamine and 1 mM pyruvate. The day before the experiment, the growth medium was replaced and cells were exposed to the uncoupler FCCP (carbonyl cyanide-4-trifluoromethoxy-phenylhydrazone), a weak acid that dissipates the proton gradient by transporting H^+^ across the inner mitochondrial membrane into the matrix. FCCP concentrations were 10 µM and 15 µM for all cells derived from osteosarcoma and colon carcinoma, respectively.

### 4.2. Cell Clones

The study was performed by using scrambled and IF_1_-silenced stable GFP-negative clones to avoid the possible negative effect of GFP expression on cellular energy charge. All stable GFP-negative clones were obtained from both 143B and HCT116 parental cells through the RNA interference technique, as previously described [37]. HCT116 GFP-negative clones were obtained by transducing parental cells using retrovirus packaged with either TR30013 (scrambled) or GI325936 (shRNA) vector following the supplier’s instructions (Origene, Rockville, MD, USA) [37,65]. The IF_1_ expression levels of colon carcinoma-derived clones used in this study are shown in Appendix A, whilst the IF_1_ levels of the osteosarcoma-derived clones were previously reported [37]. Stable clones expressing a mitochondria targeted red fluorescent protein (mtRFP) were produced from parental, scrambled and IF_1_-silenced osteosarcoma cells [37,66] and used in this study to investigate mitophagy in cancer cells.

### 4.3. Proliferation of Parental Cells

143B and HCT116 cell growth was assessed after seeding and culturing 65,000 cells in complete DMEM up to 96 h. Every 24 h, cells were collected, and counted using a MUSE cell analyzer (Merck, Merck Life Science S.r.l., Milan, Italy).

### 4.4. Cell Growth after Exposure to Uncoupling Conditions

Osteosarcoma- and colon carcinoma-derived cells (150,000 and 250,000 cells, respectively) were seeded and grown in 5 mM glucose medium and after 24 h, were treated with FCCP (10 µM or 15 µM for osteosarcoma- and colon carcinoma -derived cells, respectively). After 24 h of incubation with FCCP, all cell types were detached and centrifugated, and the same initial number of alive cells only were re-seeded and cultured in 25 mM glucose medium. The cellular growth was followed till 96 h. At 72 h and 96 h, cellular pellets were detached and incubated with Muse Count and Viability Assay Kit reagents (Luminex, Prodotti Gianni S.r.l., Milan, Italy) according to the manufacturer’s instructions. Stained samples were then acquired with the Muse cytometer set to the optimal flow rate and cut-off.

### 4.5. Cell Viability

The cell viability in 25 mM and 5 mM glucose medium was evaluated in controls and IF_1_-silenced cells in the absence/presence of FCCP (10 and 15 μM) for 24 h by cytometry using the Muse Count and Viability Assay Kit (Luminex, Prodotti Gianni S.r.l., Milan, Italy). The cells were detached, pooled with the corresponding culture medium and centrifuged. The cellular pellet was resuspended and incubated in the kit reagents according to the manufacturer’s instructions. Stained samples were then acquired with the Muse cytometer set to the optimal flow rate and cut-off.

### 4.6. SDS-PAGE and Western Blot Analysis

Cells were solubilized in RIPA buffer and the protein concentrations were quantified with the Lowry method [67,68]. Cellular lysates were separated by SDS-PAGE in Bolt 4–12% Bis-Tris Plus Gels (Thermo Fisher Scientific, Life Technologies Italia, Monza, Italy) and transferred onto nitrocellulose membranes to perform semiquantitative analysis of proteins, as previously described [69]. Chemiluminesce detection was performed with the ECL Western Blotting Detection Reagent Kit Amershan (GE Healthcare, Merck Life Science S.r.l., Milan, Italy) using the ChemiDoc MP system equipped with ImageLab software (Bio-Rad, Bio-Rad Laboratories S.r.l., Milan, Italy) to perform the densitometric scanning and analysis of the relative protein intensity.

### 4.7. Flow Cytometry Assay of Mitochondrial Membrane Potential

Time-course monitoring of mitochondrial membrane potential was performed by flow cytometry in cells loaded with the tetramethylrhodamine methyl ester (TMRM, Molecular Probes, Life Technologies Italia, Monza, Italy), a cell-permeant fluorescent lipophilic dye that accumulates in mitochondria in a Δψ_m_-dependent manner [70]. Briefly, cells were detached, counted and resuspended at the optimal concentration in 20 nM TMRM in DMEM serum free medium supplemented with substrates. After 30 min of incubation at 37 °C, the TMRM fluorescence intensity corresponding to the basal Δψ_m_ was acquired by Muse Cell Analyzer (Merck, Merck Life Science S.r.l., Milan, Italy) with λ of excitation at 532 nm and emissions collected via a 576/28 nm filter. Subsequently, TMRM fluorescence decay was monitored during the time after the incubation with 1 µM or 1.5 µM FCCP for 143B and HCT116, respectively, until a plateau. Because of the interference between FCCP and FBS, the FCCP concentrations were 10-fold reduced for the experiments performed without FBS. The F_1_F_0_-ATPase contribution to the Δψ_m_ reached in uncoupling conditions was then evaluated by adding 1.5 µM oligomycin to the sample and periodically acquiring the TMRM fluorescence until the achievement of a new stationary state. The data were analyzed using Flowing software 3.1 (Perttu Terho, Turku Bioscience, Turku, Finland), and mean fluorescence intensities ± coefficient of variation (CV) were plotted against time.

### 4.8. Fluorescence Microscopy Evaluation of Mitochondrial Membrane Potential

143B, HCT116 and related IF_1_-silenced cells grown in 25 mM glucose medium were treated or not with FCCP (10 µM or 15 µM for osteosarcoma- and colon carcinoma-derived cells, respectively) for 15 min. and 24 h. After the incubation time, cells were loaded with 20 nM TMRM for 30 min. at 37 °C, keeping FCCP. Fluorescence images were acquired using a fluorescence-inverted microscope equipped with a CCD camera (Olympus IX50, Evident Europe GmbH, Milan, Italy)). Multiple high-power images (magnification 40×) were captured with IAS2000 software (Delta Sistemi, Roma, Italy). Fluorescence micrographs of TMRM-loaded cells were obtained using specific filters with an excitation 540/20 and emission 610/40.

### 4.9. ADP/ATP Ratio Assay

The energetic charge of cells subjected to 24 h of FCCP incubation was estimated by assessing the intracellular ADP/ATP ratio through the luminometric method (Merck, Merck Life Science S.r.l., Milano, Italy). Briefly, cells were seeded in 25 mM glucose medium and after 48 h, were treated with FCCP (10 µM or 15 µM for osteosarcoma- and colon carcinoma-derived cells, respectively) for 24 h. Then, the cells were detached and resuspended in HBSS to the final concentration of 300 cells/µL. The ADP/ATP ratio of all cell types was assayed using the kit (Merck, Merck Life Science S.r.l., Milano, Italy) following the manufacturer’s instructions. The luminescence was recorded with a Luminoskan TL Plus (Labsystems, Milan, Italy) luminometer and the ADP/ATP ratio was calculated. For each uncoupled cell line, data were represented as a percentage of the ADP/ATP ratio percentage relative to the untreated condition.

### 4.10. Fluorescence Microscopy Evaluation of Mitophagy Activation

Control and IF_1_-silenced osteosarcoma clones stably expressing the mtRFP were transiently transfected with the pCMV6-LC3-YFP vector for the LC3 autophagic marker over-expression. The pCMV6-LC3-YFP vector, expressing the LC3 (microtubule-associated protein 1A/1B-light chain 3) fused to a yellow fluorescent protein (LC3-YFP), was a generous gift from Dr Lodovica Vergani, Dept. of Neurosciences, University of Padova. Briefly, the cells were transfected with 2 µg of pCMV6-LC3-YFP vector and polyethylenimine (PEI) in 1:8 DNA:PEI ratio (w/w). The day after transfection, LC3-YFP expression was checked by fluorescence microscopy and subsequently, the cells were detached, counted and seeded for the experiment. After 24 h of culture, 25 µM Chloroquine (CQ) was added to both FCCP-treated and untreated cells to prevent fusion of autophagosomes with lysosomes. Positive controls were set up by co-treating cells with CQ and 10 µM deferoxamine (DFO). After 24 h of incubation, fluorescence images of both mtRFP and LC3-YFP channels were acquired by a fluorescence-inverted microscope equipped with a CCD camera Olympus IX50 (magnification 40x; excitation 480/30 nm and emission 530/30 nm were used for LC3-YFP fluorescence detection; excitation 540/20 nm and emission 610/40 nm were used for mtRPF fluorescence detection). Fluorescence images were processed and merged by using the image deconvolution software AutoDeblur & Autovisualize 9.3 (AutoQuant Imaging Inc., Asia Imaging, 12 Marina Boulevard, Singapore).

### 4.11. Reagents

Antibodies: IF_1_ (AB110277) 1:1000, TOM20 (AB186735) 1:2000, VDAC1 (AB14734) 1:2000, LC3B (AB48394) 1:1000, ATP synthase α-subunit (AB110273) 1:1000, and ATP synthase d-subunit (AB110275) 1:5000 were from Abcam (Prodotti Gianni S.r.l., Milan, Italy); SIRT1 (8469S) 1:1000, and PGC-1α (2178S) 1:500 were from Cell Signalling Technologies (Euroclone S.p.A., Milan, Italy); PINK1 (BC100-494) 1:500 was from Novus Biological (DBA Italia, Milan, Italy); and β-actin (A5441) 1:10,000 was from Merck (Merck Life Science S.r.l., Milano, Italy). The antibodies were utilized according to the manufacturer’s instructions. All the primary antibodies were incubated overnight. Horseradish peroxidase-conjugated secondary antibodies and goat anti-mouse IgG H + L (G21040) 1:2000 were obtained from Thermo Fisher Scientific (Life Technologies Italia, Monza, Italy) and goat anti-rabbit IgG (1706515) 1:2000 was from Merk (Merck Life Science S.r.l., Milano, Italy); the secondary antibodies were incubated for 1 h. TMRM (T668) (stock solution 100 μM in DMSO and working solution in medium) was obtained from Molecular Probes (Life Technologies Italia, Monza, Italy). FCCP (C2920) (stock solution 39.3 mM in ethanol and working solution in medium), ADP/ATP ratio assay kit (MAK135), Oligomycin A (75351) (stock solution 6.32 mM in DMSO and working solution in medium), Deferoxamine (D9533) (stock solution 50 mM in H_2_O and working solution in medium), and Chloroquine (C6628) (stock solution 10 mM in H_2_O and working solution in medium) were from Merck (Merck Life Science S.r.l., Milano, Italy). The MUSE Count and Viability Assay Kit (MCH100102) was from Luminex (Prodotti Gianni S.r.l., Milan, Italy). ECL PRIME (RPN2232) or SELECT (RPN2235) were from Amersham (GE Healthcare, Merck Life Science S.r.l., Milano, Italy).

### 4.12. Statistical Analysis

All numerical data are expressed as mean ± SEM, as indicated. The number of biological replicates in independent experiments is detailed in each figure legend. The unpaired student’s *t* test or One sample *t* test were used as indicated in the legend. One sample *t* test was used for statistical analysis The one sample *t* test is a statistical hypothesis test used to determine whether an unknown population mean is different from a specific value (100% or 1 in our figures). Statistical analysis was performed using specific Excel tools by Microsoft Office. Differences were considered statistically significant for *p* < 0.05.

## Figures and Tables

**Figure 1 ijms-24-14624-f001:**
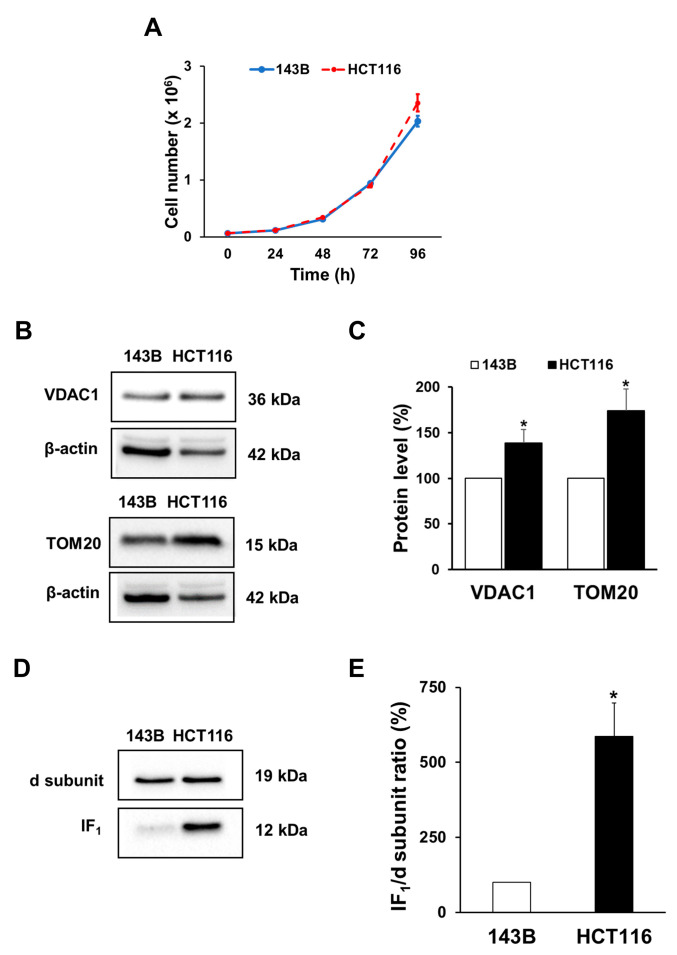
Cell growth, mitochondrial mass and IF_1_ levels in 143B and HCT116 cells. (**A**) Growth of 143B and HCT116 cells in 25 mM glucose medium. Values are means ± SEM (*n* = 3 biological replicates). (**B**,**C**) Representative immunodetection of VDAC1 and TOM20 proteins and relative densitometric analysis. Values are means ± SEM (*n* = 7 for TOM20 and *n* = 9 for VDAC1 biological replicates) and are expressed as a percentage of the 143B protein content. (**D**,**E**) Representative immunodetection of IF_1_ and ATP synthase d subunit proteins and analysis of their densitometric ratio. Values are means ± SEM (*n* = 3 biological replicates), and are expressed as a percentage of the 143B ratio. * *p* < 0.05 indicates the statistical significance of values compared with 143B cells assessed by one sample *t* test.

**Figure 2 ijms-24-14624-f002:**
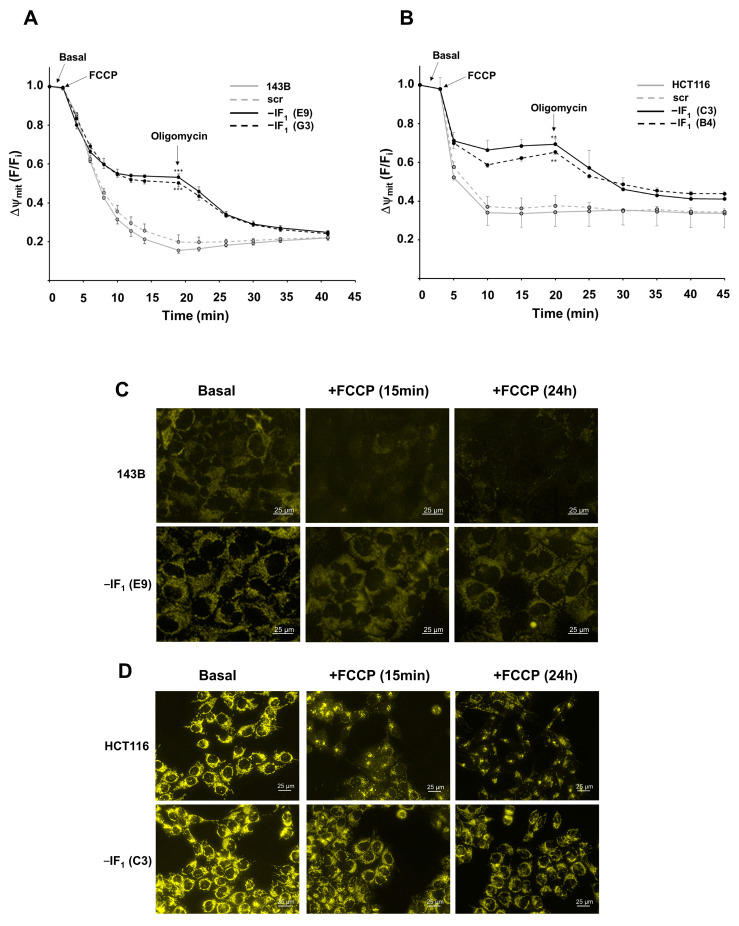
FCCP-induced mitochondrial membrane potential decay in controls and IF_1_-silenced cells. (**A**,**B**) Flow cytometry detection of Δψ_m_ changes induced in TMRM-loaded cells by the addition of FCCP (1 µM for osteosarcoma-derived cells and 1.5 µM for colon carcinoma-derived cells, respectively) and subsequently of the ATP synthase specific inhibitor oligomycin (1.5 µM). Values are means ± SEM (*n* = 3 biological replicates). ** *p* < 0.01, *** *p* < 0.005 indicates the statistical significance of values compared with relative control cells, as assessed by two-tailed *t* test. (**C**,**D**) Representative fluorescence images of IF_1_-expressing and IF_1_-silenced cells (magnification 40×) grown under basal conditions and in the presence of FCCP (1 µM for osteosarcoma-derived cells and 1.5 µM for colon carcinoma-derived cells, respectively). Between 4 and 6 images, with about 30 cells for each image, were acquired for each experimental condition.

**Figure 3 ijms-24-14624-f003:**
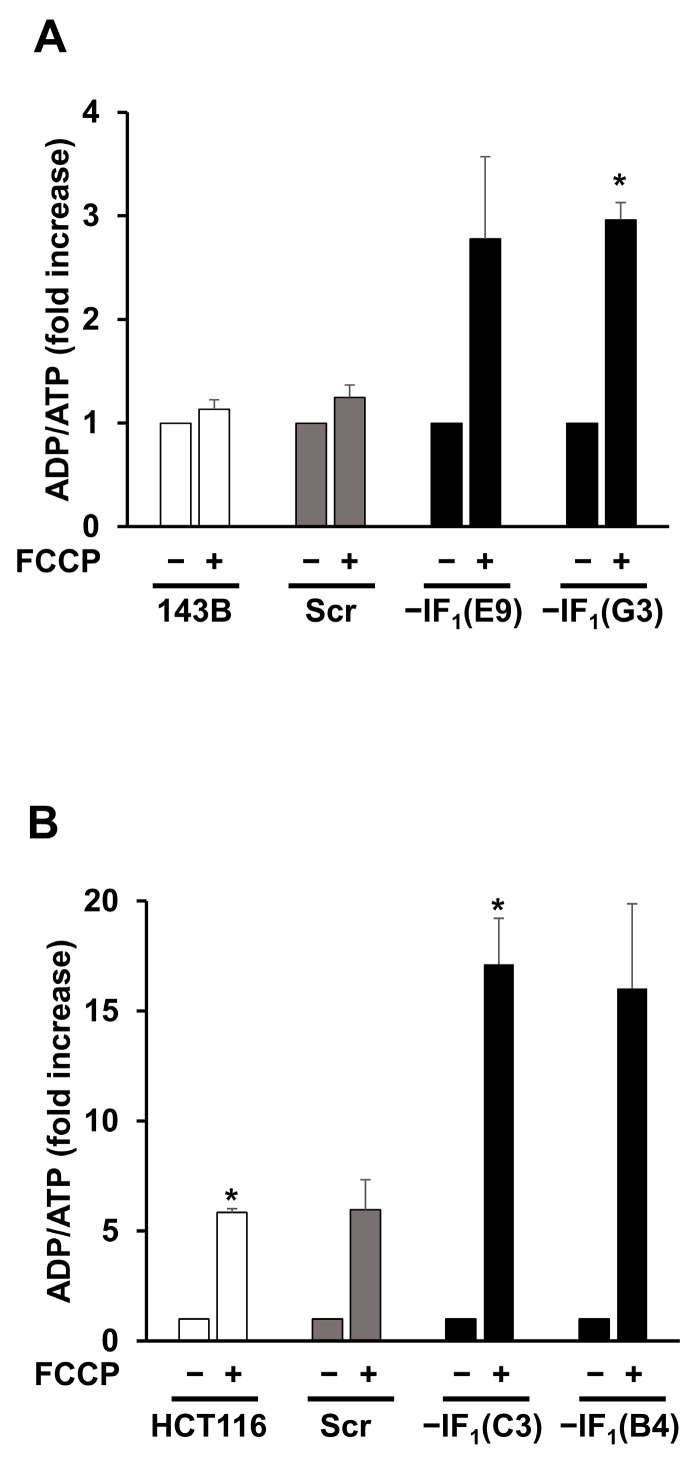
Energy charge of IF_1_-silenced and control cells under uncoupling conditions. Assessment of the ADP/ATP ratio in osteosarcoma-derived cells (**A**) and colon carcinoma-derived cells (**B**) exposed or not to 10 µM or 15 µM FCCP for 24 h, respectively. Values are reported as a percentage of untreated cells. Values are means ± SEM (*n* = 3 biological replicates). * *p* < 0.05 indicates the statistical significance of values compared with untreated cells, as assessed by one sample *t* test.

**Figure 4 ijms-24-14624-f004:**
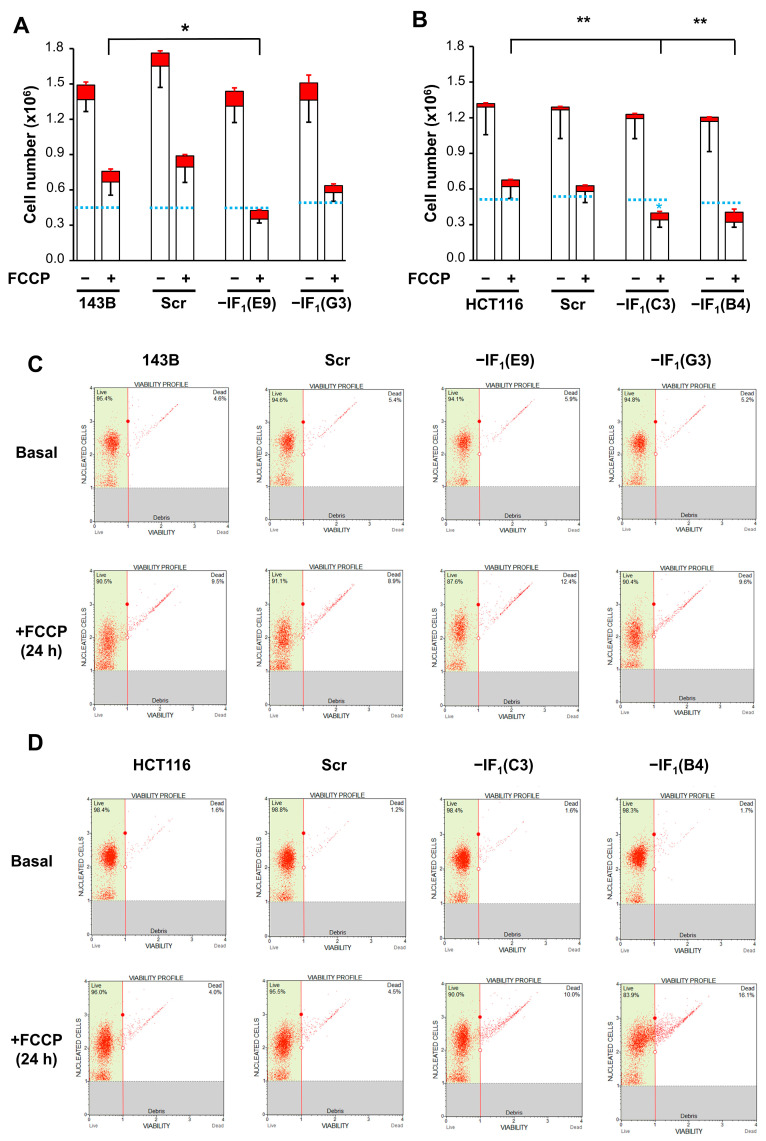
Growth and viability of uncoupled IF_1_-silenced and control cells in high glucose medium. Growth and viability of osteosarcoma-derived cells (**A**) and colon carcinoma-derived cells (**B**) maintained in 25 mM glucose medium in the presence or absence of FCCP (10 µM or 15 µM, respectively) for 24 h. Values are means ± SEM (*n* = 3 biological replicates). The red and the white boxes of the histogram represent the number of dead and live cells, respectively. Red and black bars represent the SEM of dead and live cells, respectively. * *p* < 0.05 and ** *p* < 0.01 in black indicate the statistical significance of values of live cells compared with control in uncoupling conditions, as assessed by two-tailed *t* test. * *p* < 0.05 in light blue indicates the statistical significance of cell number compared with seeded cells for each cell type (light blue dashed line), as assessed by two-tailed *t* test. (**C**,**D**) Representative cytometric analysis of viability of both controls and IF_1_-silenced cells after FCCP exposure. Dot plots with angled marker providing data on live and dead cells are shown.

**Figure 5 ijms-24-14624-f005:**
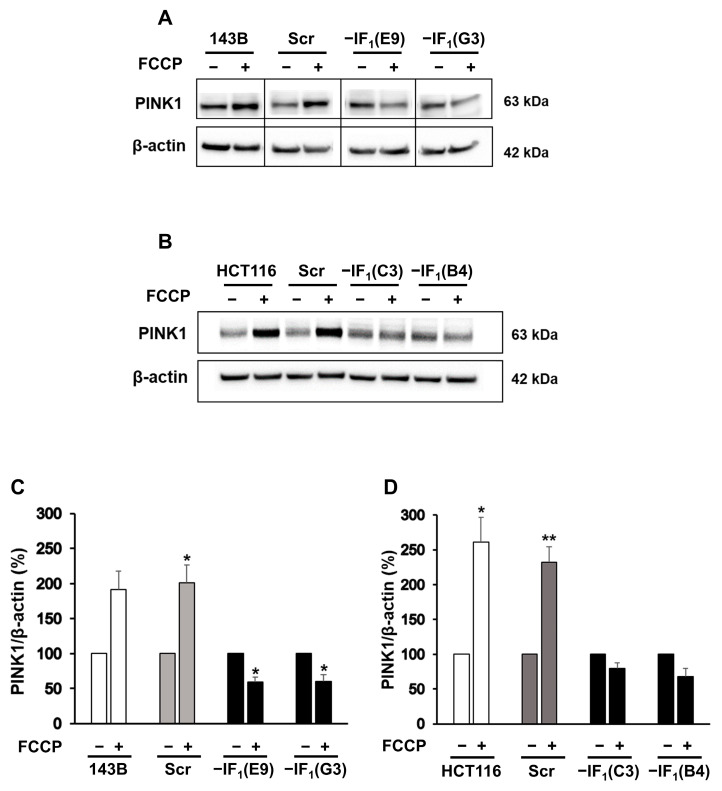
PINK1 protein levels of uncoupled IF_1_-silenced and control cells. Representative immunodetection of PINK1 in osteosarcoma-derived cells (**A**) and colon carcinoma-derived cells (**B**) exposed or not to FCCP (10 µM or 15 µM, respectively) for 24 h. Densitometric analysis of PINK1 normalized to actin (**C**,**D**). Values were reported as a percentage of the protein content relative to untreated condition for each cell type. Values are means ± SEM (*n* = 3 for osteosarcoma-derived cells and n = 4 for colon carcinoma-derived cells biological replicates). * *p* < 0.05, ** *p* <0.01 indicates the statistical significance of values compared with untreated condition, as assessed by one sample *t* test.

**Figure 6 ijms-24-14624-f006:**
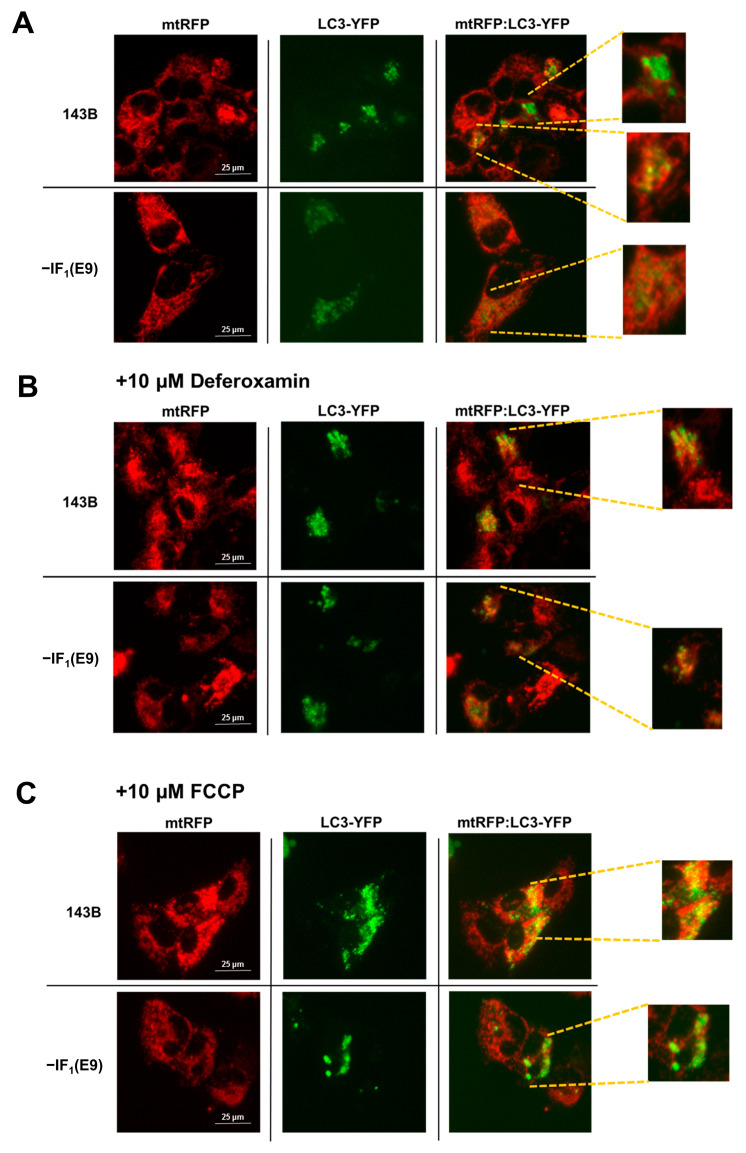
Mitophagy evaluation in uncoupled osteosarcoma cells. Representative images of parental and IF_1_-silenced osteosarcoma cells in (**A**) basal conditions; (**B**) presence of 10 µM DFO for 24 h; and (**C**) presence of 10 µM FCCP for 24 h (Magnification 40×). The inset at the right shows an enlarged detail of the mtRFP and LC3-YFP florescence co-localization. Between 4 and 6 images, with about 10 cells for each image, were acquired for each experimental condition.

**Figure 7 ijms-24-14624-f007:**
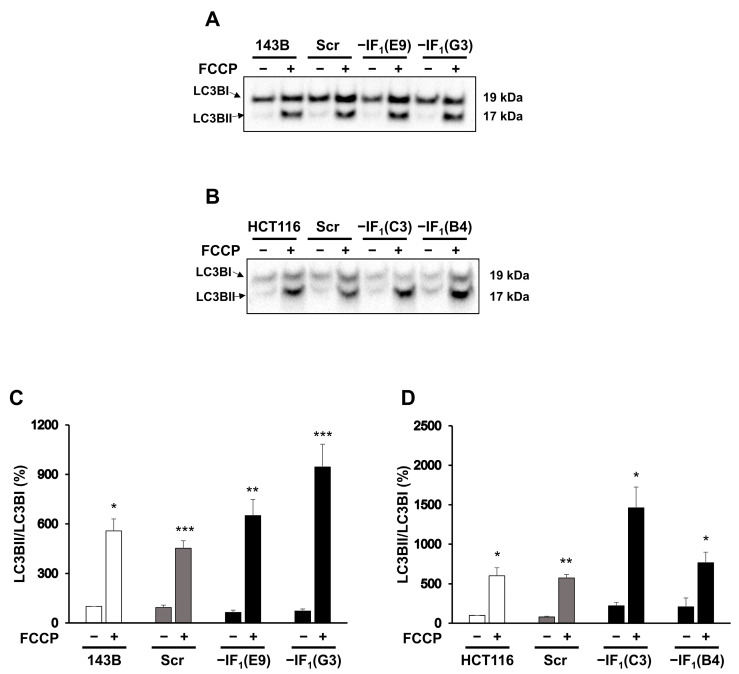
LC3B protein levels of uncoupled IF_1_-silenced and control cells. Representative immunodetection of LC3BI and II in osteosarcoma-derived cells (**A**) and colon carcinoma-derived cells (**B**) exposed or not to FCCP (10 µM or 15 µM, respectively) for 24 h. Densitometric analysis of LC3BII/I ratio (**C**,**D**). Values are reported as a percentage of the LC3BII/I ratio relative to untreated condition of the parental cell lines. Values are means ± SEM (*n* = 4 biological replicates for 143B and *n* = 3 biological replicates for HCT116). * *p* < 0.05, ** *p* <0.01, *** *p* <0.005 indicates the statistical significance of values compared with untreated condition, as assessed by one sample *t* test.

**Figure 8 ijms-24-14624-f008:**
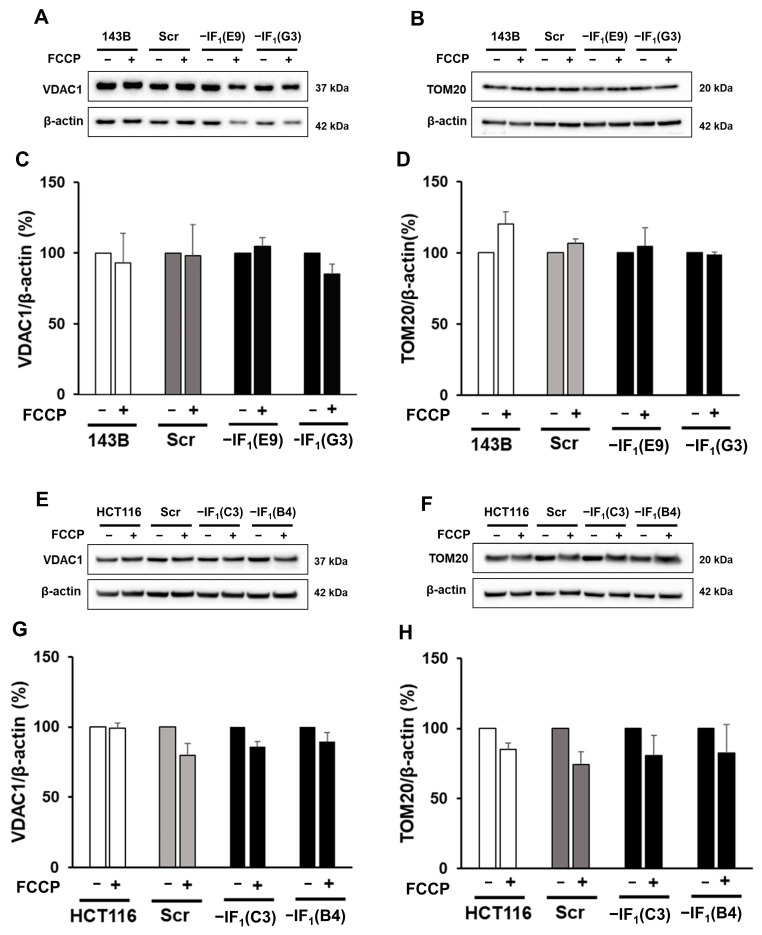
VDAC1 and TOM20 protein levels of uncoupled IF_1_-silenced and control cells. (**A**,**B**) Representative immunodetection of VDAC1 and TOM20 proteins in osteosarcoma-derived cells exposed or not to 10 µM FCCP for 24 h. (**C**,**D**) Densitometric analysis of VDAC1 and TOM20 normalized to actin and reported as a percentage of the protein content relative to untreated condition for each cell type. (**E**,**F**) Representative immunodetection of VDAC1 and TOM20 proteins in colon carcinoma-derived cells exposed or not to 15 µM FCCP for 24 h. (**G**,**H**) Densitometric analysis of VDAC1 and TOM20 normalized to actin. Values were reported as a percentage of the protein content relative to untreated condition for each cell type. Values are means ± SEM (VDAC1: *n* = 3 for osteosarcoma-derived cells, *n* = 4 for colon carcinoma-derived cells biological replicates; TOM20: *n* = 3 biological replicates).

**Figure 9 ijms-24-14624-f009:**
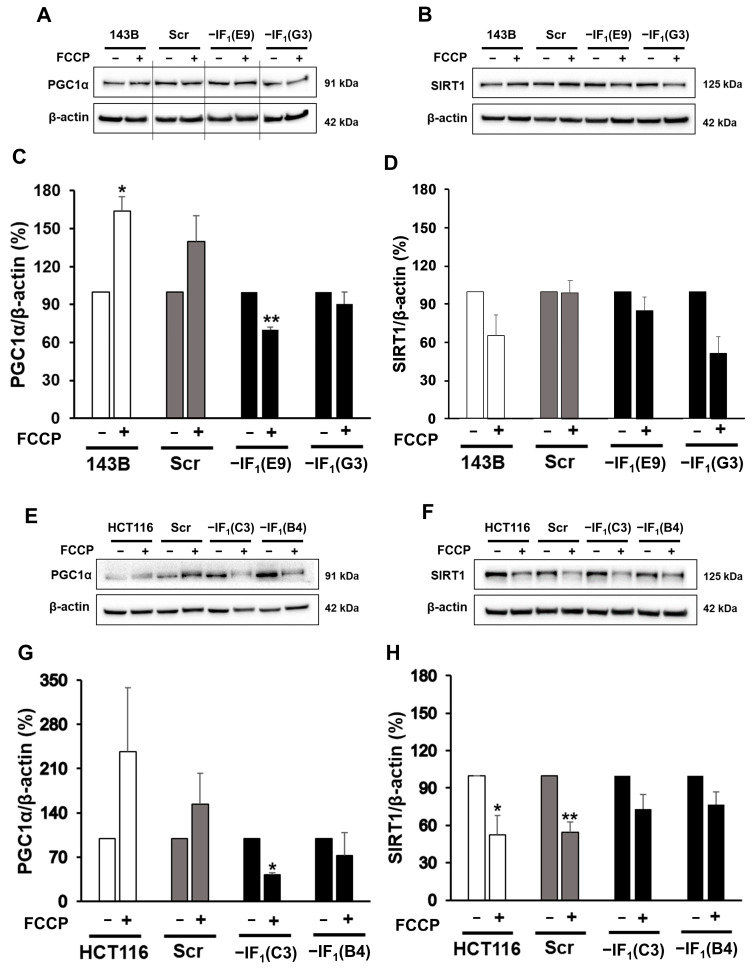
PGC1-α and SIRT1 protein levels in uncoupled IF_1_-silenced and control cells. (**A**,**B**) Representative immunodetection of PGC1-α and SIRT1 proteins in osteosarcoma-derived cells exposed or not to 10 µM FCCP for 24 h. (**C**,**D**) Densitometric analysis of PGC1-α and SIRT1 normalized to actin and reported as a percentage of the protein content relative to untreated condition for each cell type. Values are means ± SEM (*n* = 3 biological replicates). (**E**,**F**) Representative immunodetection of PGC1-α and SIRT1 proteins in colon carcinoma-derived cells exposed or not to 15 µM FCCP for 24 h. (**G**,**H**) Densitometric analysis of PGC1-α and SIRT1 normalized to actin and reported as a percentage of the protein content relative to untreated condition for each cell type. Values are means ± SEM (PGC1-α *n* = 3 and SIRT1 *n* = 5 biological replicates). * *p* < 0.05 and ** *p* < 0.01 indicate the statistical significance of values compared with untreated condition, as assessed by one sample *t* test.

**Figure 10 ijms-24-14624-f010:**
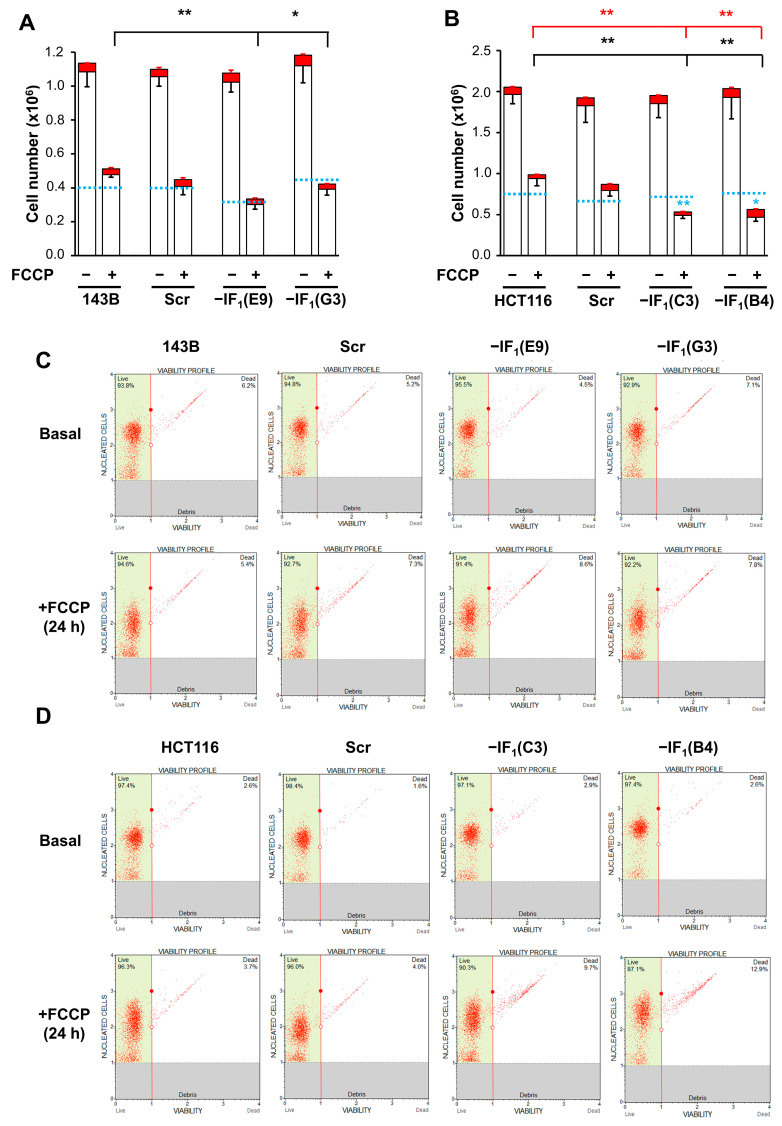
Growth and viability of uncoupled IF_1_-silenced and control cells in low-glucose medium. Growth and viability of osteosarcoma-derived cells (**A**) and colon carcinoma-derived cells (**B**) maintained in 5 mM glucose medium in the presence or absence of FCCP (10 µM or 15 µM, respectively) for 24 h. Values are means ± SEM (*n* = 3 biological replicates). The red and the white boxes of the histogram represent the number of dead and live cells, respectively. Red and black bars represent the SEM of dead and live cells, respectively. * *p* < 0.05 and ** *p* < 0.01 indicate the statistical significance of live (black) or dead (red) cells compared with their parental cells under uncoupling conditions, as assessed by two-tailed *t* test. * *p* < 0.05 and ** *p* < 0.01 in light blue indicate the statistical significance of cell number compared with seeded cells for each cell type (light blue dashed line), as assessed by two-tailed *t* test. (**C**,**D**) Representative cytometric analysis of viability of both controls and IF_1_-silenced cells after FCCP exposure. Dot plots with angled marker providing data on live and dead cells are shown.

**Figure 11 ijms-24-14624-f011:**
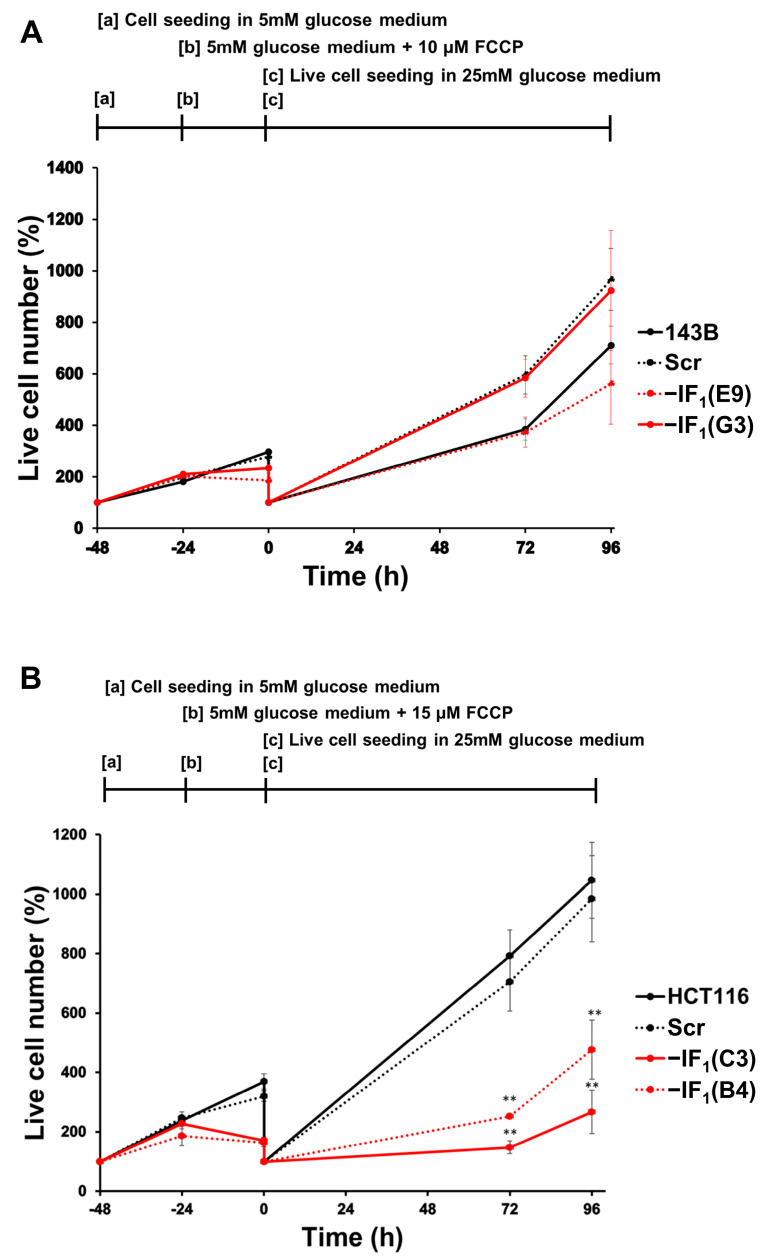
Cell proliferation of IF_1_-silenced and control cells under re-oxygenation-mimicking condition. (**A**,**B**) Osteosarcoma-derived and colon carcinoma-derived cells were seeded in 5 mM glucose medium for 24 h (a), exposed to FCCP (10 µM or 15 µM, respectively) for 24 h (b), and live cells re-seeded and grown in high glucose and FCCP-free medium up to 96 h (c). Values are reported as a percentage of cells relative to cell seeding. Values are means ± SEM (*n* = 3 biological replicates). ** *p* < 0.01 indicates the statistical significance of values of IF_1_-silenced live cells compared with controls at the same time point, as assessed by two-tailed *t* test.

**Figure 12 ijms-24-14624-f012:**
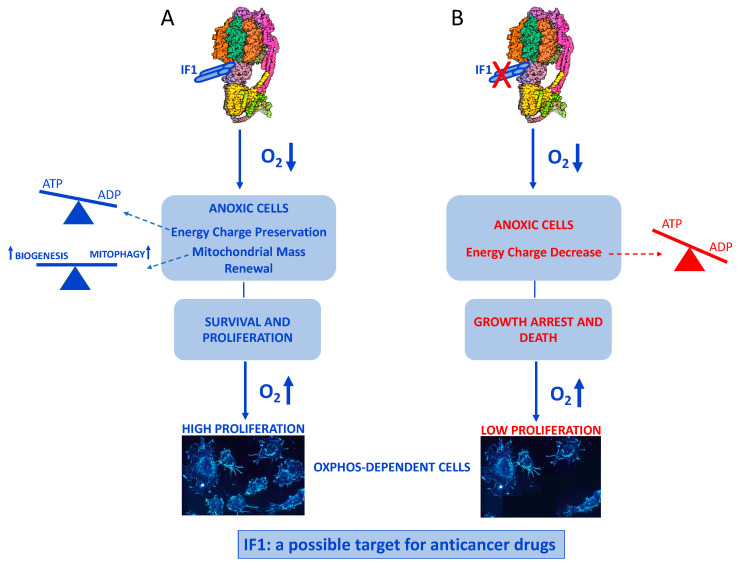
Schematic Representative cartoon of the protection exerted by IF_1_ in anoxic and re-oxygenated cancer cells, according to the present study. (**A**) Biochemical mechanisms at the basis of IF_1_ action leading to survival/proliferation of anoxic and re-oxygenated cells. (**B**) IF_1_ downregulation promotes growth arrest/death of anoxic and re-oxygenated cells. The imbalance between ATP and ADP represents the change induced by anoxia to the condition of cellular homeostasis. ATP synthase structure by PDB 6J51 (PDB DOI: 10.2210/pdb6J5I/pdb) with modifications.

## Data Availability

The data presented in this study are available on request from the corresponding author.

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
