# Peer review of "The Pro-Oncogenic Protein IF_1_ Promotes Proliferation of Anoxic Cancer Cells during Re-Oxygenation"

_ijms, 2023, doi:10.3390/ijms241914624_

Round 1

Reviewer 1 Report

In the study titled “The pro-oncogenic protein IF1 promotes proliferation of anoxic cancer cells during re-oxygenation”, the authors delve into the significance of IF1 in influencing the survival, growth, and bioenergetic responses of cancer cells, particularly in scenarios reminiscent of ischemia and reperfusion. The research centered on the effects of IF1 on osteosarcoma and colon carcinoma cells subjected to conditions mimicking ischemia and reperfusion. Cells, both with silenced IF1 and those unmodified, were subjected to the FCCP uncoupler, leading to the collapse of ΔμH+ and subsequently verifying the bioenergetics of these cellular models. Observations revealed that all cell types, irrespective of their IF1 expression status, retained their mitochondrial mass post FCCP uncoupler exposure. Interestingly, the mechanisms employed to sustain this mass varied: cells expressing IF1 showed a marked rise in mitophagy and mitochondrial biogenesis. Additionally, the existence of IF1 bestowed a proliferative edge to cells predominantly dependent on oxidative phosphorylation during simulated re-oxygenation events. The topic is intriguing, however, there are still several comments that need to be addressed.

The bands on the western blot were excessively cropped; please make the necessary adjustments.

The variation in certain experimental results seems very small, given the authors' assertion of conducting three biological replicates. The authors should clarify the type of replicates they utilized—whether they are biological or technical?

Line 573, 4.11 Reagents: It would enhance the clarity and reproducibility of the study if the authors could provide more detailed information regarding the antibodies, reagents, and kits utilized.

Line 584, 4.12 Statistical analysis. The student’s t-test can be one-tailed or two-tailed and can be applied to paired or independent samples. Please clarify which one was used. You mentioned that the t-test was used "unless otherwise indicated." It would be beneficial to mention (at least briefly) what those other tests might be and under what conditions they were used. In addition, mentioning the software used for statistical analyses can help in understanding and replicating the research.

In Figures 1B-E, where are the housekeeping proteins indicated? How have the authors assessed the quality and compared the target expression levels between 143B and HCT116 cells without normalization?

In Figures 4A and 4B, what do the white and red boxes represent? They should be clearly labeled and described in the figure legends.

In Figure 7B, where can LC3I be located? In Figures 7C and 7D, autophagy levels are typically represented by the LC3II/LC3I ratio, not the LC3B/housekeeping gene ratio.

There are numerous spelling errors and mistakes; please correct them before resubmitting. For example,

Line 460, “CO2”, the 2 should be subscripted

Line 461, “ml” should be “mL”

The anguage should be improved.

Author Response

  • The bands on the western blot were excessively cropped; please make the necessary adjustments.

We thank the reviewer for the comment. Accordingly, we have modified the cropping of all the bands on the western blots. Furthermore, the completely un-cropped gels are present in the “Original WB” PDF file.

  • The variation in certain experimental results seems very small, given the authors' assertion of conducting three biological replicates. The authors should clarify the type of replicates they utilized—whether they are biological or technical?

Concerning the statistical analysis, as reported in section 4.12, we have now specified that the experiments were performed on biological replicates and that the number of replicates is detailed in the legend of each figure.

  • Line 573, 4.11 Reagents: It would enhance the clarity and reproducibility of the study if the authors could provide more detailed information regarding the antibodies, reagents, and kits utilized.

We have better detailed the information regarding the antibodies, reagents and kit utilized.

  • Line 584, 4.12 Statistical analysis. The student’s t-test can be one-tailed or two-tailed and can be applied to paired or independent samples. Please clarify which one was used. You mentioned that the t-test was used "unless otherwise indicated." It would be beneficial to mention (at least briefly) what those other tests might be and under what conditions they were used. In addition, mentioning the software used for statistical analyses can help in understanding and replicating the research.

We thank the reviewer for the comment and we have added the missing information about the statistical analysis performed.

  • In Figures 1B-E, where are the housekeeping proteins indicated? How have the authors assessed the quality and compared the target expression levels between 143B and HCT116 cells without normalization?

We used housekeeping proteins (actin, tubulin and GAPDH) in these gels (see the figures below) but their amount was so different between the two cell lines as to make normalization and the consequent comparison between the 2 cell lines was really difficult. We preferred to consider the percentage for this reason; the two lines were always loaded on the same gel and the same mg of protein samples were loaded.

  • In Figures 4A and 4B, what do the white and red boxes represent? They should be clearly labeled and described in the figure legends.

We thank the reviewer for the comment and we have added the missing information in the figure legend.

  • In Figure 7B, where can LC3I be located? In Figures 7C and 7D, autophagy levels are typically represented by the LC3II/LC3I ratio, not the LC3B/housekeeping gene ratio.

We thank the reviewer for the comment. We previously used 4-12% polyacrylamide gels to separate LC3B bands and it was really difficult to identify LC3BI band in HCT116 cells. Following the reviewer’s suggestion, we have now repeated the analysis on both the cell lines (143B and HCT116) by using 15% polyacrylamide gels. This changed experimental condition allowed us to detect and quantify the LC3BI and II bands to evaluate the LC3BII/I ratios. The figure 7 has been replaced.

  • There are numerous spelling errors and mistakes; please correct them before resubmitting. For example,

Line 460, “CO2”, the 2 should be subscripted

Line 461, “ml” should be “mL”

We have corrected the errors in the text.

Comments on the Quality of English Language: The language should be improved.

We have edited the English language as required.

Reviewer 2 Report

There are many inconsistent results, such as the fact of using a cell line (143B), which naturally expresses low levels of IF1. Many of the original western blots show high non-specificity, there are membranes placed upside down and upside down, the supplementary figure is not understandable and has no legend, the immunofluorescences do not present the validation of the quantifications, in figure 2 there is an effect on the 143B nuclear translocation that they do not even explain, sometimes the scramble has the effect of a silenced clone, the growth and viability representations are not adequate for a good understanding, figure 9 is not credible, they do not explain why they study SIRT1 and PGC1, presents many self-citations and what definitely makes me reject it is that the authors comment that they have made biological replicates. The authors should know, within basic statistics, that when we analyze the mean with SEM, we are dealing with technical replicates, since the standard error gives us information about the dispersion that the mean of a sample of values ​​would have if they continued to be taken samples, that is, information about the technique. While we analyze the average with DS when dealing with biological replicates, since the DS is what gives us information about the variability of the samples.

Author Response

  • There are many inconsistent results, such as the fact of using a cell line (143B), which naturally expresses low levels of IF1.

The energetic metabolism of eukaryotic cells mainly occurs in mitochondria, therefore it is critical to consider their content when analyzing and comparing different types of cells. We chose two cancer cell lines containing different mitochondria mass and different IF1/ATP synthase ratio to represent cancer cells that show a rather different energetic metabolism. We believed these differences could influence the behavior of the cells under conditions mimicking low oxygen pressure and possibly their subsequent exposure to conditions re-establishing normal proton motive force. Indeed, the final goal of the study is to investigate the mechanism(s) underlying the pro-oncogenic role of this protein to get information of interest on a broad set of tumors.

  • Many of the original western blots show high non-specificity

In the “original WB” PDF file there are the images of the original and uncropped gels. The bands indicated with the arrow are those analyzed in the indicated figure. The other bands are not “high non-specificity” but other proteins, either analyzed in other figures of the manuscript or not analyzed in this manuscript. The only antibody that showed some non-specificity in 143B cell lines is PINK Ab from Novus Biological (some non-specificity is present also in the datasheet), but the immunodetected protein band chosen corresponds to the molecular weight of the protein we wanted to identify. In HCT cell lines the antibody works better than in 143B lines, showing a clear detection of the band with the same MW.

  • there are membranes placed upside down and upside down,

The reviewer should be clearer and indicate which ones. The membranes are not placed upside down, but some of them are horizontal flipped post-development to have the same gel loading order.

  • the supplementary figure is not understandable and has no legend, the immunofluorescences do not present the validation of the quantifications,

We do not understand the reviewer’s comment. The only supplemental figure of our manuscript is Figure S1 cited in section 4. Materials and Methods, 4.2 Cell clones, Line 482. This figure reported the HCT clones selected after IF1 silencing (Figure S1A-B) and the relative quantification (Figure S1C). The original and uncropped gels are in “Original WB” PDF file. There is no immunofluorescence. The legend is present in the figure.

  • in figure 2 there is an effect on the 143B nuclear translocation that they do not even explain,

We do not understand the reviewer’s comment. In figure 2 we presented the FCCP-induced mitochondrial membrane potential decay by flow cytometry (Figure A-B) and by fluorescence microscopy (Figure C-D). The probe is TMRM. The incubation with the uncoupler induces a membrane potential decrease and the TMRM fluorescence became weak and widespread because the probe leaves the mitochondria and diffuses into the cytosol. We do not understand why the reviewer speaks about “effect on the 143B nuclear translocation”

  • sometimes the scramble has the effect of a silenced clone,

We do not understand which experiment the reviewer is referring to. We present 8 figures with SCR and silenced IF1 clones from both 143B and HCT116 and only in figure 11A the 143B SCR distances itself from the parental and presents a trend more similar to one of the silenced clones. However, the trends of the 4 cell lines are not statistically different and this is our comment to this figure.

  • the growth and viability representations are not adequate for a good understanding,

The reviewer should be clearer and indicate what exactly is not adequate for a good understanding

  • figure 9 is not credible,

The reviewer should be clearer and indicate what exactly is not credible, otherwise it is only a lapidary and not very constructive judgment

  • they do not explain why they study SIRT1 and PGC1,

We have followed the reviewer’s suggestion and better detailed that we have studied SIRT1 and PGC1 as classical markers of mitochondrial biogenesis.

  • presents many self-citations

The data presented in this manuscript are part of a broader and older study regarding one of the research topics of our group. Therefore, we believe that self-citations are inevitable and indeed necessary for a better and complete understanding of this specific part of the research. Alongside some self-citations, we have included a rather rich bibliography of authors.

  • what definitely makes me reject it is that the authors comment that they have made biological replicates. The authors should know, within basic statistics, that when we analyze the mean with SEM, we are dealing with technical replicates, since the standard error gives us information about the dispersion that the mean of a sample of values ​​would have if they continued to be taken samples, that is, information about the technique. While we analyze the average with DS when dealing with biological replicates, since the DS is what gives us information about the variability of the samples.

From a strictly mathematical point of view, the reviewer is right in saying that SD is used for biological replicates and SEM for technical replicates. However, from a biological point of view, SEM for biological replicates is commonly accepted when working with syngeneic cell (same genetic background). The biological replicates for an IF1-silenced cell clone are 3 different preparations of the same cell line. Parental, scramble and IF1-silenced clones are syngeneic (same genetic background). Therefore, although they are biological replicates (different preparations) of the same cell lines, they could be considered as technical replicates, having the same genetic background. This is the reason why SEM is commonly accepted in the literature when working on syngeneic cell lines. The situation is different if you work for example with patient-derived fibroblasts (with the same mutations but different genetic background). In this case you should use SD.

Reviewer 3 Report

please double check the manuscript, I found a few minor errors throughout, but the work is quite solid.

Remove comments on authors line 82, just cite the reference for consistency.

minor editing 

Author Response

please double check the manuscript, I found a few minor errors throughout, but the work is quite solid.

Remove comments on authors line 82, just cite the reference for consistency.

We thank the reviewer for the comments. We have corrected the spelling errors and mistakes found in the text and we have removed the comment in line 82, leaving only the reference.

Comments on the Quality of English Language

minor editing 

We have edited the English language as required.

Round 2

Reviewer 1 Report

Thank you for the author's response. All my comments and concerns have been thoroughly addressed.

Reviewer 2 Report

no commnets, only for the editor